# DNA nanopores as artificial membrane channels for bioprotonics

Le Luo[1,9], Swathi Manda [2,9], Yunjeong Park[1,9], Busra Demir [3,4,5], Jesse Sanchez [1,6], M. P. Anantram [5], Ersin Emre Oren [3,4], Ashwin Gopinath [2] ✉ & Marco Rolandi [1,7,8] ✉

Biological membrane channels mediate information exchange between cells and facilitate molecular recognition. While tuning the shape and function of membrane channels for precision molecular sensing via de-novo routes is complex, an even more significant challenge is interfacing membrane channels with electronic devices for signal readout, which results in low efficiency of information transfer - one of the major barriers to the continued development of high-performance bioelectronic devices. To this end, we integrate membrane spanning DNA nanopores with bioprotonic contacts to create programmable, modular, and efficient artificial ion-channel interfaces. Here we show that cholesterol modified DNA nanopores spontaneously and with remarkable affinity span the lipid bilayer formed over the planar bio-protonic electrode surface and mediate proton transport across the bilayer. Using the ability to easily modify DNA nanostructures, we illustrate that this bioprotonic device can be programmed for electronic recognition of biomolecular signals such as presence of Streptavidin and the cardiac biomarker B-type natriuretic peptide, without modifying the biomolecules. We anticipate this robust interface will allow facile electronic measurement and quantification of biomolecules in a multiplexed manner.

In biological systems, communication between cells occurs via membrane proteins and ion channels that act as size-selective filters or stimulus-responsive molecular valves to either passively allow or actively control the flow of ions across the cell membrane[1]. Cellular communication often surpasses information processing in electronic devices in efficiency, regulation, and specificity[2,3]. Augmenting electronic devices with biological components can enable one to access, analyze, and respond to intercellular information via data transduction and signal transmission[1,4]. Examples include metal oxide semiconductors integrated with ATPase[5], carbon nanotubes[6,7] and silicon nanowires to sense pH[8], 2D transistors functionalized with gramicidin[9], organic electrochemical devices with membrane channels[10,11], and H⁺ selective bioprotonic devices integrated with gramicidin[12], alamethicin[12] and light sensitive rhodopsins[13,14]. Synthetic membrane channels can further increase the functionality of these devices with well-defined geometries, durability, robustness, and ease of

[1]Department of Electrical and Computer Engineering, Jack Baskin School of Engineering, University of California, Santa Cruz, Santa Cruz, CA 95064, USA. [2]Department of Mechanical Engineering, Massachusetts Institute of Technology, Cambridge, MA 02139, USA. [3]Bionanodesign Laboratory, Department of Biomedical Engineering, TOBB University of Economics and Technology, Ankara 06560, Turkey. [4]Department of Materials Science and Nanotechnology Engineering, TOBB University of Economics and Technology, Ankara 06560, Turkey. [5]Department of Electrical and Computer Engineering, University of Washington, Seattle, WA 98195, USA. [6]School of Chemical, Biological, and Environmental Engineering, Oregon State University, Corvallis, OR 97331, USA. [7]UC Santa Cruz Genomics Institute, University of California Santa Cruz, Santa Cruz, CA 95060, USA. [8]Institute for the Biology of Stem Cells, University of California Santa Cruz, Santa Cruz, CA 95064, USA. [9]These authors contributed equally: Le Luo, Swathi Manda, Yunjeong Park. ✉e-mail: agopi@mit.edu; mrolandi@ucsc.edu

modification[15,16]. Self-assembled synthetic membrane channels are particularly attractive due to their ease of fabrication[16]. To this end, Watson-Crick pairing based hybridization of single-stranded DNA (ssDNA) can rationally and in a bottom-up manner design self-assembled DNA nano-structures[17] that mimic membrane proteins with sophisticated architectures[18–21] and varied functionalities[22–24].

Here, we merge synthetic self-assembled DNA nanopores based ion-channels with H+ selective Palladium (Pd)-based electrodes to create a bioprotonic device that records and modulates H+ currents traversing across the bilayer membrane (Fig. 1). Unlike previous studies that used single channel ionic current measurements in membrane spanning nanopores[25,26], our device architecture enables biomolecular recognition as a function of ensemble measurement of the overall conductance change of the membrane that is an average over many ion-channels spread over several nanopore states. Utilizing programmable DNA nanopores, we showcase the versatility of the device in detecting specific biomolecules through distinct electronic signals, eliminating the need for additional pre-processing of the biomolecules.

## Results

### DNA Nanopore Bioprotonics

The DNA nanopore bioprotonic device comprises DNA nanopore ion channels spanning a supported lipid bilayer membrane (SLB) that is atop a Pd contact integrated with a microfluidic architecture (Fig. 1a and Supplementary Fig.1). A voltage ($V_{H+}$) between the Pd contact and the Ag/AgCl reference electrode positioned in the solution causes a current of H+ between the Pd contact and the solution depending on polarity[12,27]. As previously reported, this flow of H+ induces the electrochemical formation or dissolution of $PdH_x$ that results in a measurable current ($I_{H+}$) in the electronic circuit[12–14]. Although this strategy does not afford enough temporal nor spatial resolution to investigate individual ion channel states, we used it to measure the average change in membrane conductance due to ion channel insertion and activity.

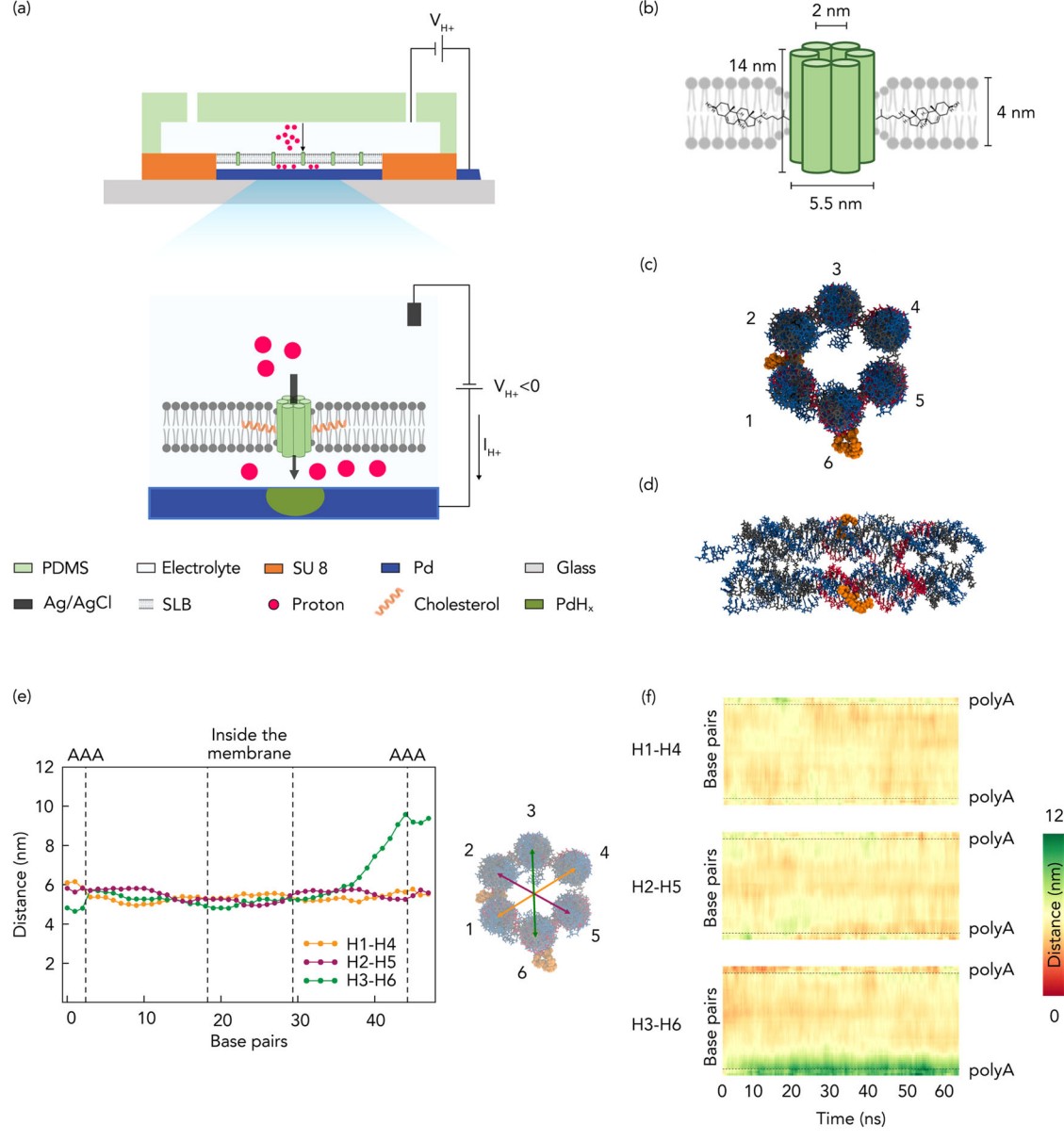

**Fig. 1 | Schematics of bioprotonic devices. a** Schematic depiction of the bioprotonic device. **b** Schematic representation of a DNA nanopore comprising six-helix bundles and 2-cholesterol anchors. **c** Top view of DNA nanopores with positioned cholesterol anchors. **d** Lateral view of DNA nanopores with positioned base pairs. **e** Simulation of the average distance between the diametrically opposite strands across the length of the nanopore. Yellow for distance between strands 1 and 4, green for distance between strands 2 and 5 and magenta for distance between strands 3 and 6. **f** Average distance heatmap for the pairs indicated in (**c**) and (**e**).

To create biomimicking ion channels that enable H⁺ transfer across the SLBs, we formed 14 nm long barrel shaped DNA nanopores via bottom-up rational design and directed self-assembly (Fig. 1b). While simple ion-channels made out of DNA duplexes that lack a central hollow pore have been previously demonstrated as effective membrane spanning ion-conduction pathways[28], we chose a DNA nanostructure geometry consisting of a central physical pore to closely mimic the membrane[29–31] and enable a larger range of signal differentiation upon varied degrees of blockage of the pore. To design the nanostructure, we adapted the single stranded tile assembly method proposed by Seeman et al.[32] to self-assemble a nanobarrel-like structure with a hollow lumen from equimolar amounts of 13 short ssDNA strands. To design the strands, we first defined the desired geometry in a hexagonal lattice-based DNA design software caDNAno[33] and filled the shape from top to bottom with an even number of parallel double helices, held together by periodic crossovers of the strands. The sequences were randomly generated and then rationally down selected to maximize primary interactions as designed and minimize secondary and tertiary complex formations. The resulting 13 ssDNA strands (Supplementary Data 1) were mixed in equimolar amounts to enable one-pot self-assembly into 6 inter-linked Helix Bundles (6HB) that form the walls of the nanopore (Fig. 1b, c, d). We functionalized the DNA nanopore with Tetri-ethylene Glycol–Cholesterol (TEG-Chol) to provide an anchor for insertion of the hydrophilic DNA nanopores into the hydrophobic environment of the SLB (Fig. 1b, c, d and Supplementary Fig. 2 and 3). Next, we conducted transmission electron microscopy (Supplementary Fig. 3), dynamic light scattering (DLS) (Supplementary Fig. 3c and d) analysis as well as molecular dynamics (MD) simulations to investigate the dimensions and the stability of the 6HB inside the SLB and its pore size under dynamic environments (Fig. 1e and f). The average distance between the diametrically opposite DNA helices across the length of the nanopore is analyzed, as depicted in Fig. 1e, providing insights into the pore size. Additionally, Fig. 1f illustrates the dynamic behavior of these distances, as they change with time on a base pair level within the DNA nanopore. Our analysis revealed that inside the membrane, the center-to-center distances of the opposite helices ranged between 5 and 6 nm. Given that the radius of a DNA double helix is 1 nm, this indicated that the average pore size fluctuated between 3 and 4 nm. Outside the membrane, the DNA helices exhibited increased mobility, resulting in some helices moving apart from one another (Fig. 1e green plot). However, as seen in the Fig. 1f, this phenomenon did not impact the stability of the pore as the TEG-Chol anchors stabilized the DNA nanopore inside the SLB. Therefore, we infer that the length of the nanopore provides sufficient area for decoration with hydrophobic anchors to enable spontaneous insertion while projecting further beyond the SLB to enable desired interactions at the lip of the nanopore without disrupting its stability within the bilayer. The small inner lumen size facilitates proton transport across the channel while obstructing proteins and other larger biomolecules to remain on the cis (negatively charged) side of the nanopore[29].

## Control of H⁺ flow with DNA nanopore bioprotonics

To validate the DNA nanopore ion-channel is indeed a H⁺ conductor, we measured the dependence of $I_{H+}$ to $V_{H+}$ in the DNA bioelectronic device (Fig. 2a). First, we verified that the bare Pd contact transfers H⁺ at the solution interface (Fig. 2a-i). To do so, we recorded $I_{H+}$ as a function of $V_{H+}$ with the following sequence as previously reported[12]. In the first step, $V_{H+} = -400$ mV for 600 s induces H⁺ to flow from the solution into the Pd contact to form $PdH_x$ (Fig. 2a–i) as indicated by $I_{H+} = -125 \pm 11$ nA (Fig. 2b). In the second step, $V_{H+} = 0$ mV transferred H⁺ from the $PdH_x$ contact into the solution[12]. Here, $I_{H+}$ indicates the prior formation of $PdH_x$ that allows H⁺ to transfer from the surface back into the solution even at $V_{H+} = 0$ mV because at a neutral pH, the protochemical potential of H⁺ in the $PdH_x$ contact is higher than the

protochemical potential of H⁺ in the solution[34,35]. We confirmed the characteristics of the Pd contact with Electrochemical Impedance Spectroscopy (EIS) (Supplementary Fig. 4 and Supplementary Table 1). Second, we confirmed that the SLBs create barriers and block H⁺ transport from the solution into the Pd surface to make sure that when we inserted the DNA nanopore we measured H⁺ transport across the nanopore (Fig. 2a-ii), as indicated by $I_{H+} = -7 \pm 1$ nA (Fig. 2b). To verify the formation of SLBs, we repeated the current measurements and reported the current which are shown as the $I_{H+}$ of 0 nM DNA in (see below) and obtained fluorescence imaging of fluorescence recovery after photobleaching measurements (FRAP) (Supplementary Fig. 5). The measured current, referred to as the leakage current, indicates that few H⁺ diffuse and leak across the bilayer membrane, possibly through the surface defects and are reduced at the Pd surface. After addition of 15 nM DNA nanopores modified with two cholesterol handles (6HB-2C) to the solution, we expected that the DNA nanopores spontaneously insert into the lipid bilayer (Fig. 2a-iii) to form membrane spanning ion channels. This insertion resulted in $I_{H+} = -52 \pm 4$ nA for $V_{H+} = -400$ mV (Fig. 2b), which is much larger than $I_{H+}$ for the SLBs coated Pd indicating that the DNA nanopores provide a pathway for H⁺ to move across the SLBs. For all measurements, to avoid the accumulation of protons on Pd contact, in the second sequence, we set the $V_{H+}$ to 0 mV. The higher photochemical potential than the electrolyte leads to the release of protons into the electrolyte and with positive $I_{H+}$. As predicted, DNA nanopores without any cholesterol handles (Fig. 2a-iv) did not insert into the SLBs as corroborated by the same observed $I_{H+}$ as recorded for the naked SLBs (Fig. 2c). Nanopores with one or three cholesterol handles (6HB-1C, 6HB-3C) (Fig. 2a-v, vi) also did not insert into the SLBs (Fig. 2c). It is likely that 6HB-1C did not insert into the SLBs because one cholesterol handle is not enough to drive the hydrophilic DNA nanopore into the hydrophobic SLBs in a membrane-spanning configuration[36]. However, with the same reasoning one would we expected to see even better insertion for 6HB-3C compared to 6HB-2C. It is likely that the increased hydrophobicity of 6HB-3C drove its aggregation in solution to minimize its interaction with water and made the hydrophobic handles unavailable for insertion into the SLB. This aggregation is confirmed by multiple bands observed for 6HB-3C in gel electrophoresis (Supplementary Fig. 3a) and a hydrodynamic radius eight times larger for 6HB-3C compared to 6HB as measured by DLS (Supplementary Fig. 3c).

## Programming the DNA nanopores for biomolecular sensing

DNA self-assembly allows for programming a desired functionality in the DNA nanopores by designing ad-hoc DNA sequences. As proof-of-concept biomolecular sensing, we programmed DNA nanopores for the detection of two proteins Streptavidin (S-avidin) and a cardiac biomarker B-type natriuretic peptide (BNP) by including a biotin handle[37] or a DNA aptamer (AP) that has been down selected using the in-vitro SELEX technology respectively on the nanopores. We did so by functionalizing 6HB-2C nanopores using two ssDNAs modified with either a Biotin handle or the AP handle at their 5' ends followed by DNA hybridization to obtain the formation of 6HB-2C-2B (Fig. 3a) and 6HB-2C-2AP (see below) nanopores respectively containing the specific tags at either end of the nano barrel. To get a clear current comparison before and after the addition of S-avidin or BNP, we increased the concentration of 6HB-2C-2B and 6HB-2C-2AP to 30 nM for the current measurements. As expected, 6HB-2C-2B nanopores inserted themselves into the SLB and resulted in a large ensemble current $I_{H+} = -96 \pm 21$ nA at $V_{H+} = -400$ mV, indicating that the nanobarrel inside the DNA nanopore aids H⁺ transport across the SLB (Fig. 3a-i and b). However, when S-avidin was introduced into the environment in five times excess concentration with respect to the nanopore concentration, the binding event of the 5 nm sized S-avidin with the biotin handle[38] on the DNA nanopore effectively occluded the nanobarrel impeding H⁺ transport across the SLB as indicated by a reduction of

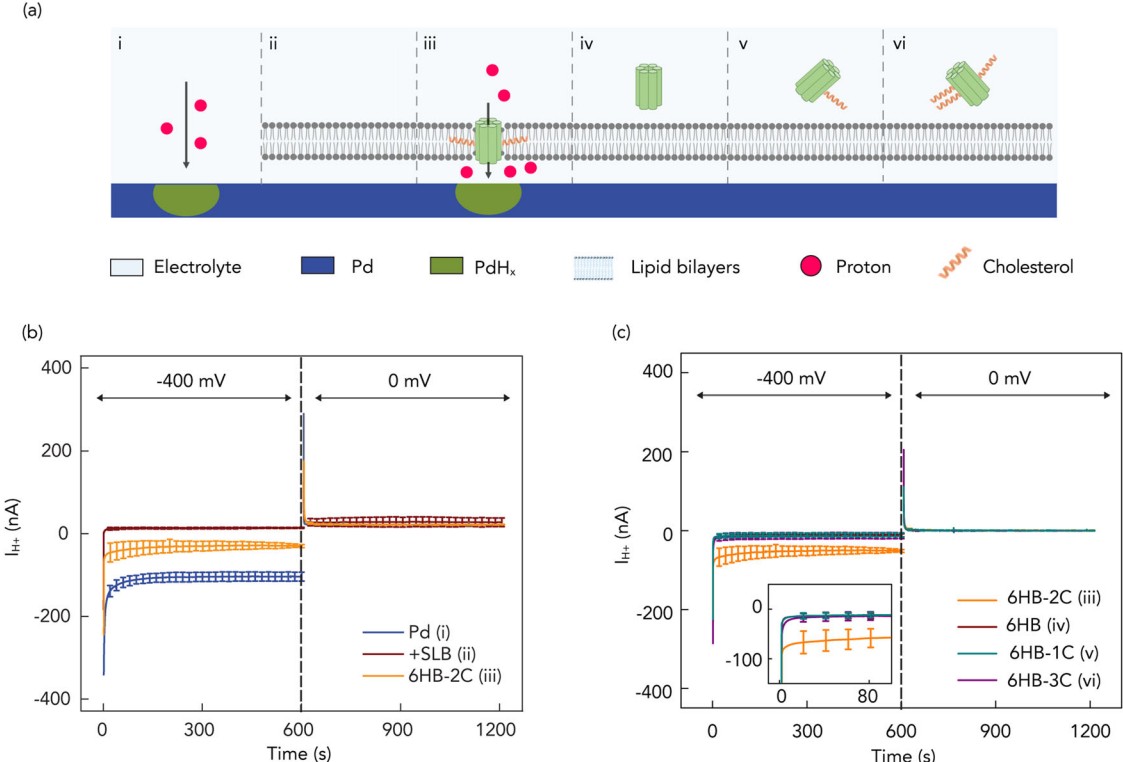

**Fig. 2 | Schematics of control of H⁺ by membrane spanning DNA nanopores. a** i) Pd contact with electrolyte solution; ii) Pd contact coated with SLBs; iii) DNA nanopores with 2-cholesterol anchors (6HB-2C); iv) DNA nanopores without cholesterol anchor (6HB); v) DNA nanopores with 1-cholesterol anchor (6HB-1C); vi) DNA nanopores with 3-cholesterol anchors (6HB-3C). **b** $I_{H+}$ versus time plot for $V = -400$ mV and $V = 0$ mV. Blue trace Pd, red trace SLB, Orange trace 6HB-2C. At $V = -400$ mV, the $I_{H+} = -125 \pm 11$ nA with bare Pd decreased to $-7 \pm 1$ nA with SLBs that indicates formed bilayers inhibit H⁺ transfer from the bulk solution to the Pd/solution interface. The $I_{H+} = -52 \pm 4$ nA with 6HB-2C confirmed that the nanopore channels support the H⁺ transport. Error bars are 1 s.d. ($n = 3$). **c** $I_{H+}$ versus time plot in different situations of Fig. 2a under $V = -400$ mV and $V = 0$ mV. Orange trace 6HB-2C (2a-iii), red trace 6HB (2a-iv), cyan trace 6HB-1C (2a-v), and purple trace 6HB-3C (2a-vi). Under $-400$ mV, we also measured $I_{H+} = -11 \pm 5$ nA, $-13 \pm 7$ nA, and $-12 \pm 4$ nA with 6HB, 6HB-1C and 6HB-3C, respectively. Error bars are 1 s.d. ($n = 3$). Only 6HB-2C provides created pathway to facilitate the flow of H⁺ to the Pd/solution interface. For all measurements, we switched the voltage to 0 mV, roughly after 600 s from the first instance of measurement, the H absorbed in Pd is oxidized to H⁺ and released back into the solution, allowing the current measured to return to 0 nA.

$I_{H+} = -12 \pm 6$ nA (Fig. 3a-ii and b) to the level of SLB leakage $I_{H+}$. This is an interesting result because prior research has shown that DNA structures without interior pores create H⁺ conduction pathways across the SLB[28] To confirm that S-avidin is indeed blocking the pore rather than plugging conduction pathways around the DNA structure, we exposed non-biotinylated 6HB-2C to the same S-avidin concentration (6HB-2C/S-avidin) in solution and did not observe appreciable change in $I_{H+} = -92 \pm 9$ nA (Fig. 3a-iii and b) because the S-avidin has no binding site available on the DNA nanopore that would result in the occlusion of the nanobarrel. Our observation and the results from ref. [37] are not necessarily contradictory because the two DNA structures are different from each other, ours contain cholesterol handles, and may interact in a different manner with the SLB. To further confirm that the current drop was indeed caused by the binding between S-avidin and biotin handle of the nanopore leading to the pathway occlusion, we also performed fluorescence imaging experiments prior to and after the addition of the proteins. To fabricate fluorescent nanopores, we modified some of the ssDNA strands with Atto 488 tags on their 5' ends (Supplementary Data 1) prior to the single pot hybridization of the nanostructures. The fluorescent images of DNA nanopores before and after the addition of S-avidin (Supplementary Fig. 6) show the same number of DNA nanopores spanning the SLBs, exhibiting that binding between S-avidin and biotin indeed blocked the channels and resulted in the decreased ensemble current rather than other possible phenomenon such as nanopore aggregation or bulk dissociation of DNA nanopores from the SLB membrane. Additionally, we measured the dependence of the $I_{H+}$ on the relative concentration of S-avidin with respect to the concentration of the biotin tagged nanopores. With the increase of S-avidin's concentration, more nanopores interacted with S-avidin leading to more blocked channels and thus a higher current decrease (Supplementary Fig. 7).

Similar experiments and controls were conducted with 6HB-2C-2AP nanopores. For the same concentration of 6HB-2C-2AP (Fig. 3c) nanopores as that of 6HB-2C-2B nanopores, the $I_{H+}$ observed at $V_{H+} = -400$ mV was $-90 \pm 3$ nA (Fig. 3c-i and d). Alike the biotin tagged nanopores, the DNA aptamer tagged nanopores inserted themselves into the SLB to form membrane-spanning ion channels and resulted in H⁺ transport across the SLB. When BNP protein was introduced into the environment in five times excess concentration with respect to the nanopore concentration, a reduced $I_{H+}$ of $-51 \pm 1$ nA was observed (Fig. 3c-ii and d) at $V_{H+} = -400$ mV. This showed that the affinity interactions between AP-BNP at the lip of the ion-channels blocked the transport of H⁺. The smaller reduction in current in the case of AP-BNP compared to the dramatic reduction observed in Biotin-S-avidin is attributed to the weaker interaction affinity ($k_D$ of $12 \pm 1.5$ nM)[39] compared against strong affinity of biotin-Streptavidin ($k_D$ of $10^{-5}$ nM)[40]. Similarly, as control, exposing non-aptamer modified 6HB-2C nanopores to the same BNP concentration in solution (6HB-2C/BNP) did not cause any appreciable change in $I_{H+}$ of $-96 \pm 9$ nA (Fig. 3c-iii and d) because the BNP has no binding site available on the DNA nanopore that would have resulted in the occlusion of the nanobarrel. By leveraging the programmability offered by the DNA nanostructures, we engineered the nanopores to demonstrate an

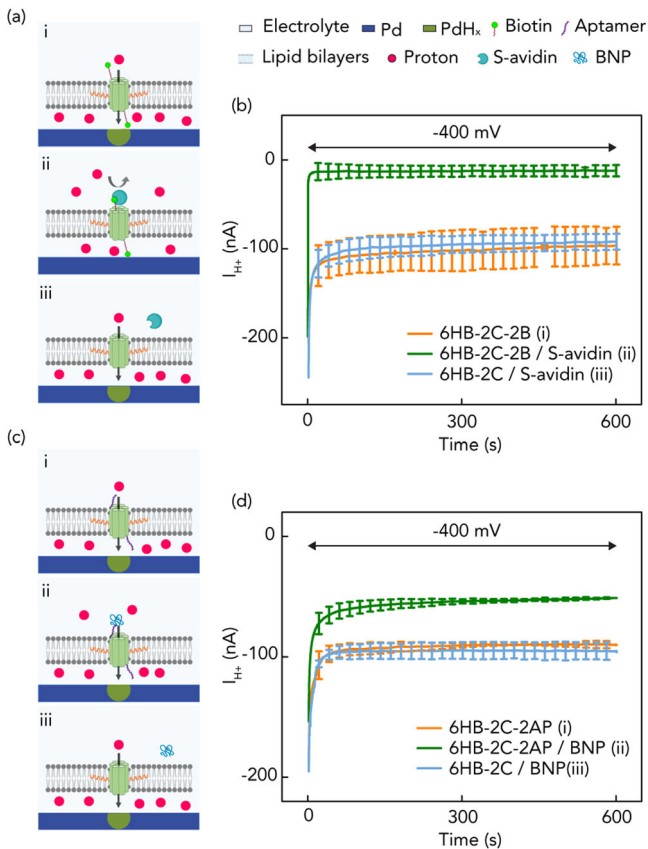

**Fig. 3 | Schematics of bioprotonic devices with biotin-Streptavidin and aptamer-peptide. a** i) 2-cholesterol handled DNA nanopores with biotin in the absence of streptavidin (6HB-2C-2B). Created pathway facilitates H⁺ transfer without inhibition of binding due to absence of streptavidin; ii) 2-cholesterol-handled DNA nanopores with binding of biotin-streptavidin (6HB-2C-2B/S-avidin). H⁺ transfer is inhibited by blocked pore channels; iii) 2-cholesterol-handled DNA nanopores without biotin in the presence of streptavidin (6HB-2C/S-avidin). The pores are not blocked by binding due to lacking biotin. **b** $I_{H+}$ versus time plot for $V = -400$ mV. Orange trace 6HB-2C-2B (3a-i), green trace 6HB-2C-2B/S-avidin (3a-ii) and blue trace 6HB-2C/S-avidin (3a-iii). We measured $I_{H+} = -96 \pm 21$ nA, $-12 \pm 6$ nA and $-92 \pm 9$ nA with 6HB-2C-2B, 6HB-2C-2B/S-avidin and 6HB-2C/S-avidin, respectively. Error bars are 1 s.d. ($n = 3$). **c** i) 2-cholesterol handled DNA nanopores with SELEX based DNA aptamer in the absence of B-type natriuretic peptide (6HB-2C-2AP). Created pathway facilitates H⁺ transfer without inhibition of binding due to absence of peptide; ii) 2-cholesterol-handled DNA nanopores with binding of aptamer-peptide (6HB-2C-2AP/BNP). H⁺ transfer is slightly inhibited by blocked pore channels; iii) 2-cholesterol-handled DNA nanopores without aptamer in the presence of peptide (6HB-2C/BNP). **d** $I_{H+}$ versus time plot for $V = -400$ mV. Orange trace 6HB-2C-2AP (3c-i), green trace 6HB-2C-2AP/BNP (3c-ii) and blue trace 6HB-2C/BNP (3c-iii). We measured $I_{H+} = -90 \pm 3$ nA, $-51 \pm 1$ nA and $-96 \pm 9$ nA with 6HB-2C-2AP, 6HB-2C-2AP/BNP and 6HB-2C/BNP, respectively. Error bars are 1 s.d. ($n = 3$).

electronic sensing response to a specific analyte in in-vitro environments without the need for modifying the analyte.

## A model for the rate constant of association and dissociation of DNA nanopores self-insertion

To better understand the dynamics of DNA nanopore insertion with the SLB, we created a model based on Langmuir's equation and absorption/desorption kinetics[41,42] to analyze the insertion process of the DNA nanopore in the lipid bilayer. In this model, we describe DNA nanopores in solution ($n$) and lipid bilayer sites where the nanopores can be absorbed ($l$) as being initially separate (Eq. 1 left side). Upon insertion of the DNA nanopore into the lipid bilayer, the DNA nanopore and the lipid bilayer sites are conjoined together and we describe this

entity as $nl$ (Eq. 1 right side).

$$n + l \overset{k_a, k_d}{\leftrightarrow} nl \tag{1}$$

The rate constant $k_a$ (M⁻¹s⁻¹) describes the absorption reaction of the DNA nanopore into the lipid bilayer and the rate constant $k_d$ (s⁻¹) describes the desorption reaction. From this model, we expect that more DNA nanopores in solution ($n$) correspond to a higher number of DNA nanopores inserted into the lipid bilayer ($nl$) resulting in $I_{H+}$ to increase as a function of DNA nanopore concentration ($C_n$) (Fig. 4a). Given the large number of DNA nanopores compared to the absorption area of the lipid bilayer, we assume $C_n$ to be constant throughout the absorption process. To fully understand the absorption and desorption kinetics, we need to now derive $k_a$ and $k_d$. To do so, we introduce the differential form of the Langmuir equation:

$$\frac{dC_{nl}}{dt} = k_a C_n C_u - k_d C_{nl} \tag{2}$$

where $C_n$ and $C_{nl}$ are the DNA nanopore concentrations in solution and lipid bilayers, respectively, and $C_u$ represents the unoccupied site concentration in the SLBs. Since $C_u$ is an unknown that we are not able to derive experimentally, we write (2) as:

$$\frac{dC_{nl}}{dt} = k_a C_n (C_{max} - C_{nl}) - k_d C_{nl} \tag{3}$$

Where $C_{max} = C_u + C_{nl}$ and $C_{max}$ is the maximum value of $C_{nl}$. We derive $C_{nl}$ by counting the number of inserted DNA nanopores ($N = C_{nl}V_lA$, $V_l$ = the volume of lipids, $A$ = Avogadro's number) as a function of $C_n$ at equilibrium using fluorescent microscopy on fluorescently tagged nanopores (Fig. 4b and Supplementary Fig. 8). Unfortunately, we are not able to measure $C_{max}$ using fluorescent microscopy for $C_n > 30$ nM because the inserted DNA nanopores are too close to each other and difficult to count. In Fig. 4a, we have shown that $I_{H+}$ increases with increasing $C_n$ for $C_n < 45$ nM and then $I_{H+}$ plateaus even if we increase $C_n$ up to 100 nM. We assume that for $C_n > 45$ nM, $C_{nl} = C_{max}$. To calculate $C_{max}$ from $I_{H+}$, we then model the DNA nanopores as resistors in parallel:

$$\frac{R_m}{N} = \frac{V_{H^+}}{I_{H^+}} \tag{4}$$

and from Eq. 4 and the slope of Fig. 4c, we calculate $R_m = 1 \times 10^{10}$ Ω and $G_m = 1/R_m = 100$ pS, the resistance and conductivity of each individual nanopore. These values are consistent with the conductivity of artificial and natural membrane channels[43,44]. Using the calculated value of $R_m$, $N_{max} = 2350$ nanopores per device and $C_{max} = 2$ nM. To conclude the derivation of $k_a$, we then observe experimentally $C_{nl}/dt$ by recording $I_{H+}$ as a function of time introducing the DNA nanopores in the solution for $t = 0$ (Fig. 4d). We can thus assume $C_{nl} = 0$ and Eq. 3 simplifies to:

$$k_a = \frac{dC_{nl}}{dt} / (C_n C_{max}) \tag{5}$$

Using $N = C_{nl}V_lA$ and Eq. 3 we can express $dC_{nl}/dt$ as:

$$\frac{dC_{nl}}{dt} = \left| \frac{dI_{H^+}}{dt} \right| \cdot \frac{R_m}{V_{H^+}} \frac{1}{V_l \times A} \tag{6}$$

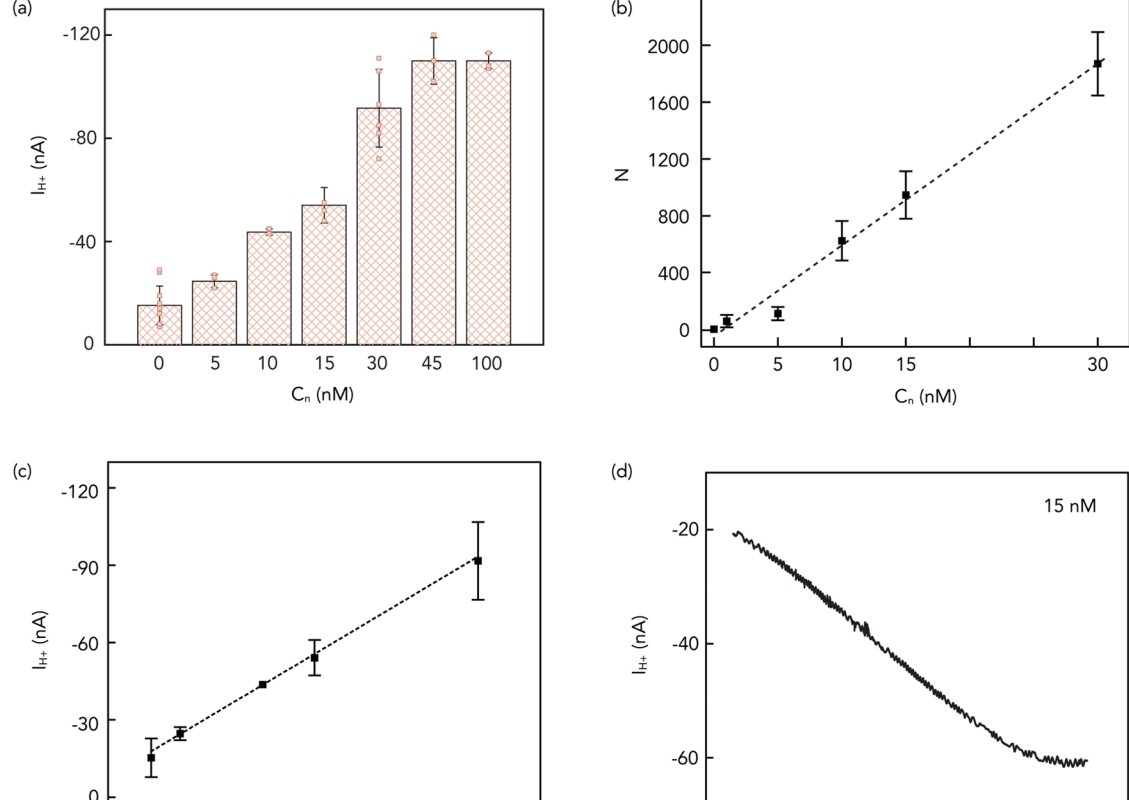

**Fig. 4 | Illustration of DNA nanopores characteristics. a** $I_{H+}$ versus the introduced 6HB-2C concentration ($C_n$) plot ($V_{H+} = -400$ mV) for 6HB-2C nanopores. We measured $I_{H+} = -15 \pm 8$ nA, $-25 \pm 3$ nA, $-44 \pm 1$ nA, $-52 \pm 4$ nA, $-92 \pm 15$ nA, $-111 \pm 9$ nA and $-110 \pm 3$ nA with 0 nM, 5 nM, 10 nM, 15 nM, 30 nM, 45 nM and 100 nM of 6HB-2C, respectively. Error bars are 1 s.d. ($n = 10, 3, 3, 6, 3$ and 3 for 0 nM, 5 nM, 10 nM, 15 nM, 30 nM, 45 nM and 100 nM respectively). **b** The number of inserted 6HB-2C nanopores ($N$) versus the introduced 6HB-2C concentration ($C_n$) plot. The value of the slope in the plot is 64. Error bars are 1 s.d. ($n = 3$). **c** $I_{H+}$ versus Number of inserted 6HB-2C nanopores ($N$) under $-400$ mV. The value of the slope in the plot is $4 \times 10^{-11}$. Error bars are 1 s.d. ($n = 3$). **d** $I_{H+}$ versus time plot during 15 nM 6HB-2C insertion process under $-400$ mV.

Combing Eqs. 5 and 6 we can express $k_a$ as:

$$k_a = \left| \frac{dI_{H^+}}{dt} \right| \cdot \frac{R_m}{V_{H^+}} \frac{1}{C_n N_{max}} \qquad (7)$$

From the slope of Fig. 4d at $t = 0$, we calculate $k_a = 8.5 \times 10^3\, M^{-1} s^{-1}$.

We then look at time $t$ when the system reaches dynamic equilibrium and $dC_{nl}/dt = 0$ and write:

$$k_a C_n (C_{max} - C_{nl,e}) = k_d C_{nl,e} \qquad (8)$$

where $C_{nl,e}$ is the adsorbate concentration in bilayers at equilibrium. We derive $C_{nl,e}$ from $I_{H+}$ and calculate $k_d = 1.9 \times 10^{-4}\, s^{-1}$. We then calculate the apparent dissociation constant to be $k_D = k_d/k_a = 22$ nM. The apparent dissociation constant indicates a high affinity of the 6HB-2C to the SLBs and is higher than the affinity of most protein-ligand interactions (100 μM–100 nM)[43,45,46].

## Discussion

We have successfully demonstrated a programmable bio-protonic device with membrane-spanning DNA nanopore ion channels as molecularly precise interconnects, which measure and control the $H^+$ transfer across the lipid bilayer interface. Leveraging the programmability of DNA constructs to custom design the nanopores and modify their surfaces, we introduced a class of self-assembling membrane-spanning molecular signal transducers. These are able to interface with bio-protonic contacts to electronically sense specific biomolecules in-vitro, bypassing the need for additional preprocessing of the biomolecules. With our device architecture, we demonstrated that the ensemble electronic current signals instead of single channel recordings can be effectively used for the electronic recognition of specific biomolecules. This approach enables the simultaneous collection of responses from multiple channels, which could potentially yield more reliable and accurate information about the target. Our ensemble method compensates for any variability or outliers in individual channel recordings, resulting in data that is more consistent and reliable, thereby enhancing the robustness and reliability in sensing the targets. In addition, this strategy greatly simplifies the device fabrication process and the recording of signals by eliminating the necessity for high precision equipment and individual tailoring associated with single-molecule devices. Furthermore, we provided valuable insights into the kinetics of DNA nanopores by conducting ensemble experiments and developing a dynamic model. These findings lay the foundations to explore potential future applications of this DNA nanopore architecture in the field of biosensing.

## Methods
### Materials
1,2-dioleoyl-sn-glycerol-3-phosphocholine (DOPC, Avanti Polar Lipids), 1,2-dipalmitoyl-sn-glycerol-3-phosphoethanolamine-N-(lissamine rhodamine B sulfonyl) (Fluorescent liposomes, Avanti Polar Lipids) were used as received for formation of supported lipid bilayers. Unmodified

ssDNA oligos in 25 nmole scale with standard purification, 3' TEG-Chol modified ssDNA oligos in 100 nmole scale with HPLC purification, 5'Bn modified ssDNA oligos in 25 nmole scale with standard purification, AP-modified ssDNA oligos in 100 nmole scale with PAGE purification and 5'Atto modified ssDNA oligos in 100 nmole scale with HPLC purification were obtained from Integrated DNA Technologies. For sequences, refer to Supplementary Data 1. The recombinant human BNP protein (ab87200) was purchased from Abcam and Streptavidin was purchased from Thermo Fisher Scientific. TE buffer 10× (pH = 8.0), $MgCl_2 \cdot 6H_2O$, 3-aminopropyl-triethoxy-silane (APTEs) and PBS (pH = 7.5) were purchased from Sigma-Aldrich. The Ag/AgCl reference electrode (RE) and counter electrode (CE) were from Warner Instruments. Glass wafers, 4-in diameter, were obtained from University Wafer Inc.

## Device architecture, fabrication and characterization
Bioprotonic devices were fabricated with conventional soft- and photo-lithography on a 500 μm thick layer of glass. The SU-8 insulating channel is 10 μm thick and the PDMS microfluidic channel is 100 μm thick on each chip. The Pd contacts, which served as electrodes for our tests, have a contact area of $0.25 \, mm^2$ (500 × 500 μm) with a thickness of 100 nm for significant interfacing with lipid solution. Pd was deposited on top of 5 nm Cr adhesion layer via electron beam evaporation. The microfluidic channel confines the flow of liquid to the top of the Pd contact and provides space to insert RE and CE (Supplementary Fig. 1).

## Electrical measurements
We used a potential cut-off of −400 mV for proton current measurement via Autolab. In the first step, we applied −400 mV for 600 seconds to induce $H^+$ to flow from the solution into the Pd contact to form $PdH_x$ and measured the proton current $I_{H+}$. In the second step, $V_{H+}$ = 0 mV transferred $H^+$ from the $PdH_x$ contact into the solution to show that at a neutral pH, the protochemical potential of $H^+$ in the $PdH_x$ contact is higher than the protochemical potential of $H^+$ in the solution.

EIS measurements were performed with Autolab, recording impedance spectra in the frequency range between 0.1 Hz–100 kHz. An AC voltage of 0.01 V and a DC voltage of 0 V versus OCP (open circuit potential) were applied (Supplementary Fig. 4 and Supplementary Table 1).

## SLB formation
DOPC liposomes were extracted and dried from a vial containing DOPC and chloroform via using nitrogen to blow it. And thus, the vial was put into a vacuum chamber for at least 6 h to dry DOPC extremely. Followingly, PBS buffer solution (pH = 7.5) was added into the vial for rehydration with the exact density (1 mg ml⁻¹). Sonication and vortex promote dissolution of DOPC in buffer solution and then 220 nm sterilizing filters purchased from Millex confine the size of vesicles. Before the deposition of SLBs on Pd contacts, the surface was hydrophilized by oxygen plasma[47]. The vesicle solution was introduced and dispensed in the microfluidic channel and the device was gently agitated for at least 8 h in high relative humidity (~95% RH) to ensure vesicle fusion[48,49] and the SLB formation, followed by rinsing with buffer solution to wash away vesicle residue that was unfused. Essentially, the SLBs mimic cell membranes, electrically insulate the Pd contact and divide the solution into two volumes: SLBs are not in direct contact with the surface of the solid substrate because of a very thin hydration layer of 1–2 nm thickness between Pd contact and SLBs on this cis-side[50]. The separation offered by this thin layer facilitates the insertion of ion channels, such as the DNA nanopores by supplying lubrication and mobility to the SLBs[51].

## Fluorescence imaging and fluorescence recovery after photobleaching measurements (FRAP)
The formation and quality of SLBs are validated by Fluorescence Recovery After Photobleaching (FRAP) (Supplementary Fig. 5), where fluorescent liposomes were combined with the DOPC lipids to form the bilayer. The samples were flushed with PBS buffer several times to remove excess fluorophores. Fluorescence imaging and FRAP experiments were performed on confocal microscopy (Leica, SP5 Confocal Microscope) with a 63× water immersion objective. DNA nanopores tagged with 488 Atto fluorophores on either end of the nano barrel were used for the fluorescence imaging experiments. 488 nm Ar laser was used for fluorescence imaging, and 543 nm and 594 nm HeNe laser was used for photobleaching. A 20 μm diameter spot in the supported lipid bilayer was photobleached, and its fluorescence intensity recovery was monitored for 30 min. The fluorescence intensity of diameter and changes over time were analyzed with Image J and fitted using a Gaussian function[52]. The diffusion coefficient was calculated with the below equation:

$$D = \frac{R_n^2 + R_e^2}{T_{1/2}}$$

where $R_n$ is the nominal radius from the user defined spot, $R_e$ is the effective radius from the bleached radius right after the bleaching process, $T_{1/2}$ is half time to recovery and the diffusion coefficient was $8.52 \, \mu m^2/sec$.

## DNA nanopores folding and characterization
The 6HB-2C DNA nanopore was assembled by heating and cooling an equimolar mixture of 11 unmodified and 2 TEG-Chol modified DNA strands (for sequences, see Supplementary Data 1). 10 μL of each of 1 μM ssDNA were mixed along with 6 μL of 200 mM MgCl₂, 10 μL of 10× TE (pH = 8.0) and MQ water to prepare a 100 μL folding mixture. The mixture was divided into 50 μL aliquots so that the solution maintains an even contact with the heating elements of the thermocycler. They were first heated to a temperature of 95 °C and then sequentially cooled to 16 °C by reducing the temperature at a rate of 0.13 °C per minute. For 6HB control nanopores without cholesterol anchors (6HB) and other variations such as 6HB-1C, and 6HB-3C, 6HB-2B, 6HB-2C, 6HB-2C-2B, 6HB-2C-2AP and fluorescent tagged nanopores, the sequences were appropriately modified (for sequences, see Supplementary Data 1).

The self-assembled structures were then characterized to confirm the correct and successful formation of the DNA nanopore. Since the structures were formed from equimolar ratios of ssDNA strands, purification was not necessary. The concentration of the resulting double stranded DNA nanostructures was analyzed with a spectrophotometer using UV absorbance spectra. Native gel electrophoresis was performed to verify the completeness of the folded structure and to verify the migration of the control nanopores without any cholesterol vs. migration of 6HB-1C, 6HB-2C, and 6HB-3C nanopores (Supplementary Fig. 3a). The 6HB-2C nanopores yielded a band, which migrated to the similar height as a control nanopore without any cholesterol anchors (Supplementary Fig. 3a, lanes 3 and 5 respectively, main band migrating at 300 bp marker). Furthermore, DLS established the monomeric nature of the nanobarrels, as only a single peak (volume-based size distribution) with an average hydrodynamic radius of 9.73 nm was observed (blue curve in Supplementary Fig. 3c). Intensity based size-correlograms (Supplementary Fig. 3d) and the polydispersity index values (Supplementary Table 2) for the 6HB and 6HB-3C nanopores show presence of multiple-size distributions containing possible higher-order aggregates since unpurified samples were used directly for the experiments. For biotin modified nanopores, 6HB-2B

and 6HB-2B-2C, the gel electrophoresis showed the accessibility of the biotin tags as slower migration patterns and dimer/quadrate aggregation patterns were observed in presence of excess Streptavidin protein (1x:20x concentration ratios) (Supplementary Fig. 3b lanes 6 and 8). No such migration pattern changes were observed in non-biotinylated nanopores, showing they were appropriate as controls (Supplementary Fig. 3b lanes 7).

## Simulation

We perform molecular dynamics (MD) simulations with NAMD software[53,54] using periodic boundary conditions. 6HB DNA nanopore design is generated with caDNAno and converted into all atom structures using automated conversation program which is available at the nanoHub web site. We covalently bound cholesterol-TEG (chol-TEG) extensions to 3'ends of designed staple strands by using the "patches" provided in the NAMD tutorial[55]. We, then, inserted the chol-TEG conjugated 6HB DNA nanopore into the pre-equilibrated DOPC lipid bilayer membrane using the CHARMM-GUI website[56]. CHARMM 36[57] and CGenFF[58] force fields were used to define chol-TEG conjugated DNA nanostructure. We placed the whole system inside 0.15 KCl electrolyte after removing overlapping lipid and water molecules. For water molecules and ions TIP3P[59] force field was used. After generating the initial system, we minimized the energy of lipid molecules for 50,000 steps by keeping chol-TEG conjugated DNA nanostructure fixed. Next, we minimized the energy of the whole system while keeping the chol-TEG conjugated DNA nanopore harmonically restrained using the exponent of two for the harmonic constraint energy function for another 50,000 steps. We released all the harmonic constraints and equilibrated the whole system for 3 ns prior MD production runs. Finally, the whole system was simulated for 64 ns at 295 K with a 2 fs timestep by saving the coordinates at every 4 fs. During the simulations, the VDW cutoff value is taken to be 12 Å. Electrostatic interactions are computed using the PME method[60], and the SHAKE algorithm is applied to keep H bonds rigid.

## Statistical & Reproducibility

In this study, current measurements were conducted at least three times independently, and the results presented here are representative of these repeated experiments. The statistical analysis was done by origin software and Microsoft Excel. The sample size for all experiments was not predetermined but was kept consistent across all trials. No data were excluded from the analysis. The experiments were not randomized.

Imaging was conducted more than three times independently, and the results presented here are representative of these repeated experiments. Each image was analyzed using Image J, and statistical distribution was performed using the Gaussian fitting of the Origin software. The sample size for all experiments was not predetermined but was kept consistent across all trials. No data were excluded from the analysis. The experiments were not randomized.

For the dynamic light scattering experiments, each independent sample was measured 5 times in the Malvern zetasizer instrument and the software presented the average results of all the trials for each sample. No statistical method was used to predetermine sample size or the number of repeats of the experiment, but was kept consistent across different samples. Randomization was not used and no data was excluded from the analysis.

## Reporting summary

Further information on research design is available in the Nature Portfolio Reporting Summary linked to this article.

## Data availability

Source data for figures are provided with this paper. Source data are provided with this paper.

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

## Acknowledgements

L.L., S.M., Y.P., B.D., J.S., M.P.A., E.E.O., A.G. and M.R. acknowledge the main financial support of the National Science Foundation (NSF 20-518). L.L. thanks to Dr. Tom Yuzvinsky in BSOE at University of California, Santa Cruz for his help about the fabrication of bioprotonic devices. B.D. acknowledges the support of the Hyak supercomputer system at the University of Washington and financial support from the TUBITAK International Doctoral Research Fellowship program (TUBITAK 2214-A). J.S. acknowledges the support of the University of California, Santa Cruz Research Experiences for Undergraduates (REU) in sustainable materials, which is supported by the National Science Foundation (NSF 22-601).

## Author contributions

L.L., S.M. and Y.P. contributed equally to this work. A.G., M.R. and M.P.A conceived the interface concept. S.M. and L.L. designed the experiments. S.M. designed and characterized the DNA nanopores. L.L. designed the bioprotonic devices and built the supported lipid bilayers. L.L. and J.S. built the integrated electronic devices. L.L. and Y.P. conducted and analyzed the electronic and optical measurements. L.L., Y.P. and S.M. analyzed the data and formulated the analytical model with M.R. B.D. developed and analyzed the nanopore-bilayer simulation models. A.G. and M.R. supervised the experiments. E.E.O and M.P.A supervised the nanopore simulations. L.L., S.M., Y.P., B.D., and M.R. wrote the manuscript. M.R. and A.G. edited the manuscript and all authors read the manuscript.

## Competing interests

The authors declare no competing interests.
