## [Peer Review File · Nature Communications]

REVIEWER COMMENTS

Reviewer #1 (Remarks to the Author):

The authors design a 6-helix pore made from DNA. The DNA self-assembly process is characterised by a number of different techniques. The correct assembly is confirmed by the gel, TEM and determination of the hydrodynamic diameter of the pores. The study of the aggregation in dependence of the cholesterol modifications is in line with previous work from the literature. The authors also use simulations to make sure that the nanopore can be stable. The main advance in the manuscript is that the authors concentrate on the development of a rate model that allows to extract dissociation constants. This is the most interesting part of the paper. The reason that this works lies in the construction of the bilayer membranes with well-defined geometry. The paper is well written, the data and figures are at the quality expected for publication in Nature Comm.

However, the authors should consider the following revisions:

1) Proton conduction through lipid bilayers is a well studied effect. The authors should estimate the proton conductivity they observe through their DOPC membranes in the absence of the DNA nanopores. See for example S. Veshaguri, S.M. Christensen, et al., D. Stamou Direct observation of proton pumping by a eukaryotic P-type ATPase Science, 351 (2016), pp. 1469-1473. The observed values will strengthen the conclusion that the authors have a well controlled membrane system and that the electrodes work.

2) The authors measure the average conductance of their membranes. They do not have the time resolution nor current resolution that is usually used in the nanopore field by authors like Howorka (Reference 1). The authors should clearly state this and emphasize that they are looking at the overall conductance change of the membrane and likely average over many nanopore states. Currently, this fact is not clear enough in the current form of the manuscript.

3) The key result in the paper is that the authors can reduce the current through the membrane to close to background (green curve Figure 3b). This result is very surprising as the streptavidin is not a perfect plug for the nanopores and current may flow along the outside of the DNA-lipid interface:

<https://doi.org/10.1021/acs.nanolett.6b02039>

The authors use tetravalent streptavidin. They should perform the following control measurements: 1) test if the nanopore structure can be forming tetramers due to 4 nanopores being bound to a single streptavidin. It is unclear if the two biotins can bind a single streptavidin. It could be that aggregation is the reason why they see a drop in current in 3b is that the pores aggregate and cannot insert anymore. 2) The authors could also add four biotin to the nanopore with a bit longer linkers to make sure that nanopore aggregation is not the origin of the reduction in current after addition of the streptavidin. This is a crucial experiment.

Reviewer #2 (Remarks to the Author):

Luo et al. report a nanopore device that is based on a small membrane-spanning DNA nanostructure. The synthetic DNA nanopore serves as an ion channel for H⁺ transport and the system is equipped with H⁺-selective Pd-based bioprotonic contacts, thus forming a biotic-abiotic device. The authors also demonstrate a simple mechanism of streptavidin detection, i.e. they observe a disruption in H⁺ current when the streptavidin binds to the biotin-modified channel and blocks the channel. The overall topic is very interesting, but as the insertion of cholesterol-modified DNA nanostructures into lipid membranes has been well documented in the literature (as well as the recognition of the molecules using a DNA nanopore), these reported experiments are not convincing enough for demonstrating the prospects of this method/device. The authors should demonstrate the feasibility of their device using additional experiments including ones they are mentioning in the discussion of the paper. With the completely new experiments and additional functions (i.e. experimenting with multiple different aptamers), the impact of the whole paper would significantly increase. Therefore, I am afraid that the paper cannot be published in this form. In addition, please find my detailed comments below:

- The authors use a term "DNA origami" when discussing their DNA nanostructure ("origami" appears also in the title). I find this a bit misleading as in the (conventional) DNA origami technique, the structure is folded from a single long "scaffold" strand, but here, the whole structure is formed using 13 short strands. The authors should change the naming throughout the manuscript.

- The authors could also add recent comprehensive reviews to the discussion such as [10.1093/nar/gkaa095](https://doi.org/10.1093/nar/gkaa095) and [10.1021/acscabm.0c00879](https://doi.org/10.1021/acscabm.0c00879)

- The DNA nanostructure design part is a bit vague, and the description of this relatively simple design sounds rather complicated. As mentioned above, the authors use 13 short strands to form the structure, but they call these 13 strands "aptamers". However, they do not show that these particular sequences would have any specific function. The authors should explain what is the functionality of these particular sequences if any.

- Related to the previous point: "aptamers" are also mentioned in the discussion. However, as long as such experiments are not carried out, I unfortunately find all of this hype/speculation a bit misleading.

- Overall, the discussion part is full of ideas how the DNA structure could be modified but there is no experiments carried out toward this direction. I strongly believe these scientific visions of the authors, but I also believe these experiments (or at least some of them) should be performed to increase the impact of the paper.

- The main manuscript indicates that TEG (a spacer for the cholesterol modification) is "tetra ethyl glycol" while Supplementary info says it is "Tri-ethylene Glycol". Please clarify.

- Supplementary Information Figure 3: The DLS data should also be accompanied with the polydispersity index (PDI) values as well as with the auto-correlation function fit.

- Supplementary Information Figure 4: The equivalent circuit looks quite complicated. How was the fitting to the data performed and what were the results? Could the authors also plot the real and

imaginary part of the impedance as a so-called Nyquist / Cole-Cole plot?

Reviewer #3 (Remarks to the Author):

This paper seeks to demonstrate that DNA superstructures can be designed to insert within lipid bilayer membranes and, in doing so, provide a mechanism for proton transport. The overall idea is interesting and there is a high probability that the main hypothesis is correct. There are however some dangling issues that would be good to address before acceptance to any journal.

First of all, could it be that the nanopores induce disruption of the lipid bilayer structure near the site of intercalation so that proton transport is accelerated? Avidin is sufficiently large that it could serve to “plug” these distortions. Not sure how this could be probed, but it is recommended that giant unilamellar vesicles be included in these studies, since they would probe the properties of the lipid bilayer/DNA construct (for example melting, stability, and permeability as a function of DNA loading) in the absence of an underlying substrate.

Why is the magnitude of I_{H^+} generally lower when $V_{H^+} = 0$ mV as compared to when $V_{H^+} = -400$ mV?

It is not clear to me that there are sufficient experimental details on how to design the DNA nanopores. Presumably these details are in references 21-23?

It is relevant to point out that there is no “biotic” element in this manuscript. Typically that would require a living component, for example cells. In the manuscript, the DNA nanopores mediate proton conduction across a synthetic lipid bilayer construct.

I looked and could not find widespread use of the concept of “protochemical potential”. Is this the same as the chemical potential of H^+ as a function of electrode potential and concentration?

Finally, the majority of the text in the Discussion section is dedicated to speculations on future applications and speculations that reach too far. Would be better to summarize the scientific relevance of the work, which is innovative. No need to overhype.

In sum, this is original thinking, although it is not clear to me that it describes sufficient work and controls that verify the hypothesis to warrant publication in Nature Communications in its current state.

Dear Reviewers:

Find answers attached.

Reviewer #1 (Remarks to the Author):

The authors design a 6-helix pore made from DNA. The DNA self-assembly process is characterized by a number of different techniques. The correct assembly is confirmed by the gel, TEM and determination of the hydrodynamic diameter of the pores. The study of the aggregation in dependence of the cholesterol modifications is in line with previous work from the literature. They also use simulations to make sure that the nanopore can be stable. The main advance in the manuscript is that the authors concentrate on the development of a rate model that allows to extract dissociation constants. This is the most interesting part of the paper. The reason that this works lies in the construction of the bilayer membranes with well-defined geometry. The paper is well written, the data and figures are at the quality expected for publication in Nature Comm.

Comments:

However, the authors should consider the following revisions:

1) Proton conduction through lipid bilayers is a well-studied effect. The authors should estimate the proton conductivity they observe through their DOPC membranes in the absence of the DNA nanopores. See for example S. Veshaguri, S.M. Christensen, et al., D. Stamou Direct observation of proton pumping by a eukaryotic P-type ATPase Science, 351 (2016), pp. 1469-1473. The observed values will strengthen the conclusion that the authors have a well-controlled membrane system and that the electrodes work.

>> We would like to thank you for taking the time to review our manuscript. We have measured the conductivity through the DOPC and clarified in the manuscript: "Second, we confirmed that the SLBs create a barrier and block H⁺ transport from the solution into the Pd surface as indicated by I_{H⁺} = -15 ± 8 nA (Fig. 2b). This is necessary to ensure that when we insert the DNA nanopore we measured H⁺ transport across the nanopore channels instead of across the membrane (Fig. 2a-ii). The measured current, referred to as the leakage current, indicates that few H⁺ diffuse and leak across the bilayer membrane, possibly through the surface defects and are reduced at the Pd surface."

2) The authors measure the average conductance of their membranes. They do not have the time resolution nor current resolution that is usually used in the nanopore field by authors like Howorka (Reference 1). The authors should clearly state this and emphasize that they are looking at the overall conductance change of the membrane and likely average over many nanopore states. Currently, this fact is not clear enough in the current form of the manuscript.

>> We appreciate your observation about the measurement of the average conductance of our membranes. As you have correctly pointed out, our methodology differs from what is usually used in the nanopore field and focuses on the overall conductance change of the membrane. To make this clearer, we have added the following sentence in the main

section of the manuscript: “Unlike previous studies that used single channel ionic current measurements in membrane spanning nanopores^{34,35}, our device architecture enables biomolecular recognition as a function of ensemble measurement of the overall conductance change of the membrane that is an average over many ion-channels spread over several nanopore states.”

3)The key result in the paper is that the authors can reduce the current through the membrane close to the background (green curve Figure 3b). This result is very surprising as the streptavidin is not a perfect plug for the nanopores and current may flow along the outside of the DNA-lipid interface: <https://doi.org/10.1021/acs.nanolett.6b02039>

Reply: Thank you for the comment. We have modified the manuscript as follows “ This is an interesting result because prior research has shown that DNA structures without interior pores create H⁺ conduction pathways across the SLB.³⁷ To confirm that S-avidin is indeed blocking the pore rather than plugging conduction pathways around the DNA structure, we exposed non-biotinylated 6HB-2C to the same S-avidin concentration in solution and did not observe appreciable change in I_{H⁺} (Fig. 3a-iii, b) because the S-avidin has no binding site available on the DNA nanopore that would result in the occlusion of the nanobarrel. Our observation and the results from ref. 37 are not necessarily contradictory because the two DNA structure are different from each other, ours contain cholesterol handles, and may interact in a different manner with the SLB.”

4)The authors use tetravalent streptavidin. They should perform the following control measurements: a) test if the nanopore structure can be forming tetramers due to 4 nanopores being bound to a single streptavidin. It is unclear if the two biotins can bind a single streptavidin. It could be that aggregation is the reason why they see a drop in current in 3b is that the pores aggregate and cannot insert anymore. b) The authors could also add four biotins to the nanopore with a bit longer linkers to make sure that nanopore aggregation is not the origin of the reduction in current after addition of the streptavidin. This is a crucial experiment.

Reply: Thanks for your helpful comments to further validate our use of tetravalent streptavidin. We have performed additional experiments to confirm that the interaction between streptavidin in the solution and biotin on the nanopore rather than aggregation causes the drop in current in Figure 3b. We added these results in Supplementary Figure 5, Supplementary Figure 6 and in the section “Programming the DNA nanopores for biomolecular sensing” where we clarified as follows:

“ To further confirm that the current drop was indeed caused by the binding between S-avidin and biotin handle of the nanopore leading to the pathway occlusion, we also performed fluorescence imaging experiments prior to and after the addition of the protein. To fabricate fluorescent nanopores, we modified some of the ssDNA strands with Atto 488 tags on their 5’ ends (Supplementary Table S1) prior to the single pot hybridization of the nanostructures. The fluorescent images of DNA nanopores before and after addition of S-avidin (Supplementary Fig.6) show the same number of DNA nanopores spanning the SLBs exhibiting that binding between S-avidin and biotin indeed blocked the channels and resulted in the decreased ensemble current rather than other possible

phenomenon such as nanopore aggregation or bulk dissociation of DNA nanopores from the SLB membrane. Additionally, we measured the dependence of the I_{H^+} on the relative concentration of S-avidin with respect to the concentration of the biotin tagged nanopores. With the increase of S-avidin's concentration, more nanopores interacted with S-avidin leading to more blocked channels and thus a higher current decrease (Supplementary Fig.7).”

Reviewer #2 (Remarks to the Author):

Luo et al. report a nanopore device that is based on a small membrane-spanning DNA nanostructure. The synthetic DNA nanopore serves as an ion channel for H^+ transport and the system is equipped with H^+ -selective Pd-based bioprotonic contacts, thus forming a biotic-abiotic device. The authors also demonstrate a simple mechanism of streptavidin detection, i.e. they observe a disruption in H^+ current when the streptavidin binds to the biotin-modified channel and blocks the channel. The overall topic is very interesting, but as the insertion of cholesterol-modified DNA nanostructures into lipid membranes has been well documented in the literature (as well as the recognition of the molecules using a DNA nanopore), these reported experiments are not convincing enough for demonstrating the prospects of this method/device. The authors should demonstrate the feasibility of their device using additional experiments including ones they are mentioning in the discussion of the paper. With the completely new experiments and additional functions (i.e. experimenting with multiple different aptamers), the impact of the whole paper would significantly increase. Therefore, I am afraid that the paper cannot be published in this form. In addition, please find my detailed comments below:

Comments:

- *The authors use the term "DNA origami" when discussing their DNA nanostructure ("origami" appears also in the title). I find this a bit misleading as in the (conventional) DNA origami technique, the structure is folded from a single long "scaffold" strand, but here, the whole structure is formed using 13 short strands. The authors should change the naming throughout the manuscript.*

Reply: Thanks for bringing up this terminology concern. We agreed that the term “DNA origami” typically refers to structures folded from a single, long scaffold strand. In light of this, we have replaced “DNA origami” with “DNA nanostructure” throughout the manuscript to avoid any confusion. Your attention to detail is much appreciated, and has greatly improved the clarity of our paper.

-*The authors could also add recent comprehensive reviews to the discussion such as 10.1093/nar/gkaa095 and 10.1021/acsabm.0c00879.*

Reply: Thanks for the useful reference. We already discussed related contents about related DNA technologies discussed in the reference in “DNA nanopore Bioprotonics” section.

-The DNA nanostructure design part is a bit vague, and the description of this relatively simple design sounds rather complicated. As mentioned above, the authors use 13 short strands to form the structure, but they call these 13 strands "aptamers". However, they do not show that these particular sequences would have any specific function. The authors should explain what is the functionality of these particular sequences if any.

Reply: Thank you for your insightful comments. Upon review, we understand how our use of the term "aptamer" may have led to confusion. In our current DNA nanostructure designs (6HB or 6HB-2C), these 13 short strands do not specifically serve as traditional aptamers (a specific target-binding function). To provide greater clarity and avoid any potential misinterpretation, we have revised our manuscript to refer to the oligo strands as "DNA strands" rather than "aptamers". We appreciate your careful review and valuable feedback.

- Related to the previous point: "aptamers" are also mentioned in the discussion. However, as long as such experiments are not carried out, I unfortunately find all of this hype/speculation a bit misleading.

Reply: Thank you for your feedback regarding the use of the term "aptamers" in our discussion. We understand your concern and agree that it's important to use terminology accurately to avoid confusion or misleading interpretation. Accordingly, we have revised our manuscript and replaced all instances of "aptamers" with "ssDNA oligos". This change reflects the fact that in this context, these sequences do not serve the specific binding function of aptamers. However, we did perform additional experiments where we used specific aptamers in the nanopore design. Therefore, we have updated the discussion section that is relevant to the aptamers as follows: " Variations of this next-generation bio-protonic device, such as aptamer tagged DNA nanopores will facilitate recognition and quantification of specific biomolecules. To make the nanopore respond to the presence of a molecule (DNA, RNA, proteins, or small molecule), we can design a DNA or RNA aptamer (or other affinity binders) on the cap of the nanopore as demonstrated in the experiments involving nanopores for BNP recognition. Free aptamer will remain unstructured and does not significantly affect the proton transport through the pore, whereas upon binding its target, it will undergo a significant conformational change and significantly reduced the nominal diameter of the ion channel leading to a marked reduction in the measured proton conductivity."

- Overall, the discussion part is full of ideas how the DNA structure could be modified but there are no experiments carried out toward this direction. I strongly believe these scientific visions of the authors, but I also believe these experiments (or at least some of them) should be performed to increase the impact of the paper.

Reply: Thank you for your suggestions. In response, we performed additional experiments that further illustrate these possibilities. Specifically, we conducted some

experiments to detect a cardiac marker B-type natriuretic peptide (BNP) using a SELEX technology-based aptamer in the modified nanopore design. Additionally, we experimented with different concentrations of streptavidin that could provide valuable insight into the dynamics of our system.

These data are included in the revised manuscript, Figure 3 and Supplementary Figure 7 as shown below:

“Similar experiments and controls were conducted with 6HB-2C-2AP nanopores. For the same concentration of 6HB-2C-2AP nanopores as that of 6HB-2C-2B nanopores, the steady-state I_{H^+} observed at $V_{H^+} = -400$ mV was -91 ± 4 nA (Fig. 3c-i, d). Alike the biotin tagged nanopores, the DNA aptamer modified nanopores inserted themselves into the SLB to form membrane-spanning ion-channels and resulted in H^+ transport across the SLB. When BNP protein was introduced into the environment in 5 times excess concentration with respect to the nanopore concentration, a reduced steady-state I_{H^+} of -51 ± 1 nA was observed (Fig. 3c-ii, d) at $V_{H^+} = -400$ mV. This showed that the affinity interactions between AP-BNP at the lip of the ion-channels blocked the transport of H^+ ions leading to a reduced ensemble current. The smaller reduction in current in the case of AP-BNP compared to the dramatic reduction observed in Biotin-S-avidin is attributed to the weaker interaction affinity (K_D of $12 + 1.5$ nM) compared against strong affinity of biotin-Streptavidin. Similarly, as control, exposing non-aptamer modified 6HB-2C nanopores to the same BNP concentration in solution did not cause any appreciable change in I_{H^+} (Fig. 3c-iii, d) because the BNP has no binding site available on the DNA nanopore that would have resulted in the occlusion of the nanobarrel. By leveraging the programmability offered by the DNA nanostructures, we engineered the nanopores to demonstrate an electronic sensing response to a specific analyte in in-vitro environment without the need for modifying the analyte.

Additionally, we measured the dependence of the I_{H^+} on the relative concentration of S-avidin with respect to the concentration of the biotin tagged nanopores. With the increase of S-avidin's concentration, more nanopores interacted with S-avidin leading to more blocked channels and thus a higher current decrease (Supplementary Fig.7).”

After adding new experimental results with aptamer modified nanopores, and the effect of variations in protein concentration on current drop, we believe that the impact of our paper will increase. We also modified our discussion section as follows:

“We have successfully demonstrated a programmable bio-protonic device with membrane-spanning DNA nanopore ion channels as molecularly precise interconnects that measure and control the H^+ transfer across lipid bilayer interface. Leveraging the programmability of DNA constructs for custom designing the nanopores and modifying their surfaces, we introduced a new class of self-assembling membrane spanning molecular signal transducers that can be interfaced with the bio-protonic contacts to electronically sense specific biomolecules in vitro eliminating the need for additional preprocessing of the biomolecules. With our device architecture, for the first time we demonstrated that the ensemble electronic current signals instead of single channel recordings can be used for electronic recognition of specific biomolecules thus greatly

simplifying the process to fabricate the device and record the signals. Furthermore, for the first time, we explored the kinetics of DNA nanopore experimentally through ensemble experiments. These experiments lay the foundations to explore potential future applications of this DNA nanopore architecture in biosensing.”

- *The main manuscript indicates that TEG (a spacer for the cholesterol modification) is "tetra ethyl glycol" while Supplementary info says it is "Tri-ethylene Glycol". Please clarify.*

Reply: Thank you for pointing out this inconsistency in our manuscript. We have corrected and clarified the term TEG to “Tri-ethylene Glycol” in our manuscript.

- *Supplementary Information Figure 3: The DLS data should also be accompanied with the polydispersity index (PDI) values as well as with the auto-correlation function fit.*

Reply: Thank you for your comment regarding the DLS data. We agree to providing auto-correlation functions fit and polydispersity index values for better understanding our system. Accordingly, we have included additional sub-plot Fig. 3d and Table S3 in the revised Supplementary section. The descriptions are as follows:

Supplementary Figure 3 (d): “Size correlograms for 6HB and 6HB-3C nanopores showing the raw correlation function versus delay time data in the form of $G_2(\tau) - 1$. Both the samples show multiple scattering with intercepts less than 1 owing to the fact that the intensity-based size calculations include large scattering effects from multiple-size populations and aggregates present even in extremely low fractions.”

Supplementary Table S3. Polydispersity Index (PI) values for dynamic light scattering data: “Intensity based Polydispersity index (PI) and hydrodynamic Z-average size (cumulants mean) values for 6HB and 6HB-3C nanopores as observed on Malvern zetasizer instrument. Since a small percentage of aggregates are expected and can heavily skew the calculations owing to large scattering effects, number based mean values are used and contrasted against the intensity based-calculations to provide a more relevant estimate of the nanopore population distribution.”

- *Supplementary Information Figure 4: The equivalent circuit looks quite complicated. How was the fitting to the data performed and what were the results? Could the authors also plot the real and imaginary part of the impedance as a so-called Nyquist / Cole-Cole plot?*

Reply: Thanks for your helpful comments. We agree that a visual representation of the impedance data can provide additional insight. In response to your comments, we have included the fitting data method and results in the supplementary Figure 4 and Table S2, and the Nyquist plot in the supplementary Figure 4 as below:

Supplementary Fig. 4: “EIS measurement of bioprotonic device and lipid bilayer. (a) Equivalent circuit schematic utilized to fit experimental data. (Top) Bioprotonic device, (Bottom) Bioprotonic device with SLB or SLB with DNA nanopore. The electrolyte solution resistance, R_s , in series with membrane capacitance, C_m , membrane resistance, R_m , double layer capacitance, C_{dl} , charge transfer resistance, R_{ct} , adsorption resistance R_p , and adsorption capacitance. The fitting to the experimental data was performed using the ZSimpWin software, and the results are provided in Table S1. (b) Nyquist plot illustrating the relationship between the real and imaginary part of the impedance for both bioprotonic devices and lipid bilayers. (c) Bode plot depicting the magnitude and phase of impedance as a function of frequency (Black: Pd and Red: SLB).”

Table S2. Fitted parameter for equivalent circuit model of the bioprotonic device and lipid bilayer

	R_s	C_m	R_m	CPE_{dl}	R_{ct}	CPE_p	R_p
Bare Pd	753.9			8.082e-8	2.09e5	1.328e-7	4.551e7
Pd/SLB	753.4	6.959e-6	3.953e6	1.15e-7	4.787e4	8.919e-7	1.097e6

“The overall impedance of lipid bilayer with bioprotonic device was found to be higher than that of the bare device, as evidenced by the larger semicircle in the Nyquist plot. However, in the table, the charge transfer resistance (R_{ct}) values were similar between the two systems, as shown in the fitted parameters in Table S1. This discrepancy can be attributed to the presence of a gap between the lipid bilayer and the device surface. It may result in an additional resistance component, the membrane resistance (R_m), which contributes to the overall impedance of the lipid bilayer with bioprotonic devices. In addition, the R_m value obtained in this study is similar to the reported in the reference paper, suggesting that the lipid bilayer does not significantly impede the charge transfer process of the device surface but introduces an additional resistive element.”

Reviewer #3 (Remarks to the Author):

This paper seeks to demonstrate that DNA superstructures can be designed to insert within lipid bilayer membranes and, in doing so, provide a mechanism for proton transport. The overall idea is interesting and there is a high probability that the main hypothesis is correct. There are however some dangling issues that would be good to address before acceptance to any journal.

Comments:

1. First of all, could it be that the nanopores induced disruption of the lipid bilayer structure near the site of intercalation so that proton transport is accelerated? Avidin is sufficiently large that it could serve to “plug” these distortions.

Not sure how this could be probed, but it is recommended that giant unilamellar vesicles be included in these studies, since they would probe the properties of the lipid bilayer/DNA construct (for example melting, stability, and permeability as a function of DNA loading) in the absence of an underlying substrate.

Reply: Thank you for your comments about the nanopore-induced disruption of the lipid bilayer and the potential of proton transport. In the manuscript, we conducted experiments by adding S-avidin to the system which has DNA nanopores not tagged with biotin. We did not observe the expected change in conductivity. This indicates that S-avidin alone does not have the anticipated effect on proton transport or lipid bilayer disruption. But your input has prompted us to improve with better explanations in the sentences:

“As a control, exposing non-biotinylated 6HB-2C to the same S-avidin concentration in solution did not cause any appreciable change in I_{H^+} (Fig. 3a-iii, b) because the S-avidin has no binding site available on the DNA nanopore that would result in the occlusion of the nanobarrel. To further confirm that the current drop was indeed caused by the binding between S-avidin and biotin handle of the nanopore leading to the pathway occlusion, we also performed fluorescence imaging experiments prior to and after the addition of the proteins.”

2. Why is the magnitude of I_{H^+} generally lower when $V_{H^+} = 0$ mV as compared to when $V_{H^+} = -400$ mV?

Reply: We would like to thank you for taking the time to review our manuscript. In response to your feedback, we already explained in the manuscript but we have further improved the explanation provided in the “Control of H^+ flow with DNA Nanopore Bioprotonics” section of the manuscript as below:

“To validate the DNA nanopore ion-channel is indeed a H^+ conductor, we measured the dependence of I_{H^+} to V_{H^+} in the DNA bioelectronic device (Fig. 2a). First, we verified that the bare Pd contact transfers H^+ at the solution interface (Fig. 2a-i). To do so, we recorded I_{H^+} as a function of V_{H^+} with the following sequence as previously reported²¹. In the first step, $V_{H^+} = -400$ mV for 600 seconds induces H^+ to flow from the solution into the Pd contact to form PdH_x (Fig. 2a -i) as indicated by $I_{H^+} = -131 \pm 26$ nA (Fig. 2b). In the second step, $V_{H^+} = 0$ mV transferred H^+ from the PdH_x contact into the solution²¹. Here, I_{H^+} indicates the prior formation of PdH_x that allows H^+ to transfer from the surface back into the solution even at $V_{H^+} = 0$ mV because at a neutral pH, the protochemical potential of H^+ in the PdH_x contact is higher than the protochemical potential of H^+ in the solution^{43,44}.”

3. It is not clear to me that there are sufficient experimental details on how to design the DNA nanopores. Presumably these details are in references 21-23?

Reply: Thanks for the comment. While the references do contain the details of the general DNA nanopore design and folding processes, we have added detailed description on how to design the DNA nanopores in “DNA nanopore Bioprotonics” section and in the “Methods” section under DNA nanopores folding and characterization. The added clarification is as follows:

“To create biomimicking ion channels that enable H^+ transfer across the SLBs, we formed 14 nm long barrel shaped DNA origami nanopores via bottom-up rational design and directed self-assembly (Fig. 1b). While simple ion-channels made out of DNA duplexes that lack a central hollow pore have been previously demonstrated as effective membrane

spanning ion-conduction pathways³⁷, we chose a DNA nanostructure geometry consisting of a central physical pore to closely mimic the membrane³⁸⁻⁴⁰ and enable a larger range of signal differentiation upon varied degrees of blockage of the pore. To design the nanostructure, we adapted the single stranded tile assembly method proposed by Seeman and colleagues⁴¹ to self-assemble a nanobarrel-like structure with a hollow lumen from equimolar amounts of 13 short ssDNA strands. To design the strands, we first defined the desired geometry in a hexagonal lattice-based DNA design software caDNAo⁴² and filled the shape from top to bottom with an even number of parallel double helices, held together by periodic crossovers of the strands. The sequences were randomly generated and then rationally down selected to maximize primary interactions as designed and minimize secondary and tertiary complex formations. The resulting 13 ssDNA strands (Supplementary Table S1) were mixed in equimolar amounts to enable one-pot self-assembly into 6 inter-linked Helix Bundles (6HB) that form the walls of the nanopore (Fig. 1b, c, d). We functionalized the DNA nanopore with Tetra-ethylene Glycol–Cholesterol (TEG-Chol) to provide an anchor for insertion of the hydrophilic DNA nanopores into the hydrophobic environment of the SLB (Fig. 1b, c, d and Supplementary Fig. 2 and 3).”

4. It is relevant to point out that there is no “biotic” element in this manuscript. Typically that would require a living component, for example cells. In the manuscript, the DNA nanopores mediate proton conduction across a synthetic lipid bilayer construct.

Reply: Thanks for the comment. Given that we did not experiment with or discuss any living elements in this manuscript we have removed all “biotic” descriptors or references in the revised version.

5. I looked and could not find widespread use of the concept of “protochemical potential”. Is this the same as the chemical potential of H+ as a function of electrode potential and concentration?

Reply: Thanks for the comment. To avoid confusion, we changed “protochemical” to “electrochemical”

6. Finally, the majority of the text in the Discussion section is dedicated to speculations on future applications and speculations that reach too far. Would be better to summarize the scientific relevance of the work, which is innovative. No need to overhype. In sum, this is original thinking, although it is not clear to me that it describes sufficient work and controls that verify the hypothesis to warrant publication in Nature Communications in its current state.

Reply: Thank you for your suggestions. In response, we performed additional experiments that further illustrate these possibilities. Specifically, we conducted some experiments to detect a cardiac marker B-type natriuretic peptide (BNP) using a SELEX technology-based aptamer in the modified nanopore design. Additionally, we

experimented with different concentrations of streptavidin that could provide valuable insight into the dynamics of our system.

These data are included in the revised manuscript, Figure 3 and Supplementary Figure 7 as shown below:

“Similar experiments and controls were conducted with 6HB-2C-2AP nanopores. For the same concentration of 6HB-2C-2AP nanopores as that of 6HB-2C-2B nanopores, the steady-state I_{H^+} observed at $V_{H^+} = -400$ mV was -91 ± 4 nA (Fig. 3c-i, d). Alike the biotin tagged nanopores, the DNA aptamer modified nanopores inserted themselves into the SLB to form membrane-spanning ion-channels and resulted in H^+ transport across the SLB. When BNP protein was introduced into the environment in 5 times excess concentration with respect to the nanopore concentration, a reduced steady-state I_{H^+} of -51 ± 1 nA was observed (Fig. 3c-ii, d) at $V_{H^+} = -400$ mV. This showed that the affinity interactions between AP-BNP at the lip of the ion-channels blocked the transport of H^+ ions leading to a reduced ensemble current. The smaller reduction in current in the case of AP-BNP compared to the dramatic reduction observed in Biotin-S-avidin is attributed to the weaker interaction affinity (K_D of $12 + 1.5$ nM) compared against strong affinity of biotin-Streptavidin. Similarly, as control, exposing non-aptamer modified 6HB-2C nanopores to the same BNP concentration in solution did not cause any appreciable change in I_{H^+} (Fig. 3c-iii, d) because the BNP has no binding site available on the DNA nanopore that would have resulted in the occlusion of the nanobarrel. By leveraging the programmability offered by the DNA nanostructures, we engineered the nanopores to demonstrate an electronic sensing response to a specific analyte in in-vitro environment without the need for modifying the analyte.

Additionally, we measured the dependence of the I_{H^+} on the relative concentration of S-avidin with respect to the concentration of the biotin tagged nanopores. With the increase of S-avidin's concentration, more nanopores interacted with S-avidin leading to more blocked channels and thus a higher current decrease (Supplementary Fig.7).”

And we have toned down the discussion

“We have successfully demonstrated a programmable bio-protonic device with membrane-spanning DNA nanopore ion channels as molecularly precise interconnects that measure and control the H^+ transfer across lipid bilayer interface. Leveraging the programmability of DNA constructs for custom designing the nanopores and modifying their surfaces, we introduced a new class of self-assembling membrane spanning molecular signal transducers that can be interfaced with the bio-protonic contacts to electronically sense specific biomolecules in vitro eliminating the need for additional preprocessing of the biomolecules. With our device architecture, for the first time we demonstrated that the ensemble electronic current signals instead of single channel recordings can be used for electronic recognition of specific biomolecules thus greatly simplifying the process to fabricate the device and record the signals. Furthermore, for the first time, we explored the kinetics of DNA nanopore experimentally through ensemble

experiments. These experiments lay the foundations to explore potential future applications of this DNA nanopore architecture in biosensing.”

REVIEWERS' COMMENTS

Reviewer #1 (Remarks to the Author):

The authors improved the paper. The supplementary figure with the check if tetravalent streptavidin leads to large-scale aggregation is valuable.

There is two recommendations remaining before publications:

- 1) After reading the discussion again, they may want to consider to remove some uses of "the first time".
- 2) The authors should expand the discussion a bit and explain why they believe an "ensemble" method like theirs complements or outperforms single-molecule approaches.

Reviewer #2 (Remarks to the Author):

The authors have fully addressed my concerns. The quality of the manuscript has been significantly improved through the revisions and the additional experiments. I believe the manuscript can now be published.

Reviewer #3 (Remarks to the Author):

I think that the authors have done a best faith effort to improve the clarity of the manuscript and to take all of the Reviewers' comments into account. The manuscript should be ready to publish.

Reviewer #1 (Remarks to the Author):

The authors improved the paper. The supplementary figure with the check if tetravalent streptavidin leads to large-scale aggregation is valuable.

There are two recommendations remaining before publications:

1) After reading the discussion again, they may want to consider to remove some uses of "the first time".

Reply: Thanks for the suggestion. We have removed the words "for the first time" in the discussion section.

2) The authors should expand the discussion a bit and explain why they believe an "ensemble" method like theirs complements or outperforms single-molecule approaches.

Reply: Thank you for this helpful comment. We improved the discussion section as follows:

"We have successfully demonstrated a programmable bio-protonic device with membrane-spanning DNA nanopore ion channels as molecularly precise interconnects, which measure and control the H⁺ transfer across the lipid bilayer interface. Leveraging the programmability of DNA constructs to custom design the nanopores and modify their surfaces, we introduced a novel class of self-assembling membrane-spanning molecular signal transducers. These are able to interface with bio-protonic contacts to electronically sense specific biomolecules in-vitro, bypassing the need for additional preprocessing of the biomolecules. With our device architecture, we demonstrated that the ensemble electronic current signals instead of single channel recordings can be effectively used for the electronic recognition of specific biomolecules. This approach enables the simultaneous collection of responses from multiple channels, which could potentially yield more reliable and accurate information about the target. Our ensemble method compensates for any variability or outliers in individual channel recordings, resulting in data that is more consistent and reliable, thereby enhancing the robustness and reliability in sensing the targets. In addition, this strategy greatly simplifies the device fabrication process and the recording of signals by eliminating the necessity for high precision equipment and individual tailoring associated with single-molecule devices. Furthermore, we provided valuable insights into the kinetics of DNA nanopores by conducting ensemble experiments and developing a dynamic model. These findings lay the foundations to explore potential future applications of this DNA nanopore architecture in the field of biosensing."

Reviewer #2 (Remarks to the Author):

The authors have fully addressed my concerns. The quality of the manuscript has been significantly improved through the revisions and the additional experiments. I believe the manuscript can now be published.

Reviewer #3 (Remarks to the Author):

I think that the authors have done a best faith effort to improve the clarity of the manuscript and to take all of the Reviewers' comments into account. The manuscript should be ready to publish.